# Marine carbon cycle response to a warmer Southern Ocean: the case of the Last Interglacial

Dipayan Choudhury[1], Laurie Menviel[1], Katrin J. Meissner[1,2], Nicholas K.H. Yeung[1,2], Matthew Chamberlain[3], and Tilo Ziehn[4]

[1]Climate Change Research Centre, University of New South Wales, Sydney NSW, Australia
[2]ARC Centre of Excellence for Climate Extremes, University of New South Wales, Sydney NSW, Australia
[3]CSIRO Oceans and Atmosphere, Hobart TAS, Australia
[4]CSIRO Oceans and Atmosphere, Aspendale VIC, Australia

**Correspondence:** Dipayan Choudhury (d.choudhury@unsw.edu.au)

**Abstract.** Recent studies investigating future warming scenarios have shown that the ocean carbon sink will weaken over the coming century due to ocean warming and changes in oceanic circulation. However, significant uncertainties remain regarding the magnitude of the oceanic carbon cycle response to warming. Here, we investigate the Southern Ocean's (SO, south of 40°S) carbon cycle response to warmer conditions, as simulated under Last Interglacial boundary conditions (LIG, 129-115 thousand years ago, ka). We find a ∼150% increase in carbon dioxide ($CO_2$) outgassing over the SO at the LIG compared to pre-industrial (PI), due to a 0.5°C increase in SO sea surface temperatures. This is partly compensated by an equatorward shift of the Southern Hemisphere (SH) westerlies and weaker Antarctic Bottom Water formation, which lead to an increase in dissolved inorganic carbon (DIC) in the deep ocean at the LIG compared to PI. These deep ocean DIC changes arise from increased deep and bottom water residence times, and higher remineralization rates due to higher temperatures. While our LIG simulation features a large reduction in SO sea-ice compared to PI, we find that changes in sea ice extent exert a minor control on the marine carbon cycle. The projected future strengthening and poleward shift of the SH westerlies coupled to warmer conditions at the surface of the SO should thus weaken the capacity of the SO to absorb anthropogenic $CO_2$ over the coming century.

## 1 Introduction

Future increases in atmospheric carbon dioxide ($CO_2$) concentration are unequivocally projected to further warm the Southern Ocean (SO) and reduce sea ice concentrations (Bracegirdle et al., 2020). The current state of knowledge suggests the mitigating effects of carbon cycle feedbacks on global warming to be less efficient under future scenarios (Pachauri et al., 2014). This primarily stems from the changes in carbon uptake by the terrestrial biosphere and ocean under a changing climate. Both land and ocean presently act as sinks of anthropogenic carbon, each absorbing about 25% of anthropogenic emissions (Friedlingstein et al., 2020), with 40% of the ocean sink being attributed to the SO (Caldeira and Duffy, 2000; DeVries, 2014). The SO $CO_2$ uptake weakened during the 1990s due to a strengthening and poleward shift of the Southern Hemisphere (SH) mid-latitude westerlies (Lovenduski et al., 2007; Le Quéré et al., 2007; Zickfeld et al., 2007; Gruber et al., 2019a, b), but strengthened in the

2000s due to cooling over the Pacific and increased stratification over the Atlantic and Indian sectors of the SO (Landschuetzer et al., 2016; Gruber et al., 2019b).

In addition, the consensus amongst studies analysing future climate simulations points towards amplified warming over high latitudes (Holland and Bitz, 2003; Smith et al., 2019; Fan et al., 2020). For example, the SO annual mean sea surface temperature (SST) anomaly is projected to exceed 0.5°C for Shared Socioeconomic Pathway scenario (SSP) 245 and 1.5°C for SSP 585 at 2100 relative to 2015 (Bracegirdle et al., 2020; Tonelli et al., 2021), with reduced sea ice during austral spring (Roach et al., 2020). At the same time, SH westerlies are projected to strengthen and shift poleward over the coming century

(Collins et al., 2013; Zheng et al., 2013; Goyal et al., 2021). Overall, climate change will reduce the amount of anthropogenic carbon taken up by the ocean (Plattner et al., 2001; Bernardello et al., 2014; Wang et al., 2014; Howes et al., 2015) by reducing $CO_2$ solubility (Bernardello et al., 2014), weakening the efficiency of the biological pump (Boyd, 2015), increasing outgassing from upwelling of Circumpolar Deep Waters (CDW) due to stronger and poleward shifted SH westerlies (Lovenduski et al., 2007; Zheng et al., 2013; Gruber et al., 2019a), and reducing sea-ice extent (Shadwick et al., 2021). However, uncertainties

still remain in regards to the response of the SO carbon cycle under a warmer climate.

    The Last Interglacial (LIG, 129-115 thousand years ago, ka) was the warmest interglacial of the last 800 thousand years (Masson-Delmotte et al., 2013; PAGES, 2016). The warmer climate at the LIG is primarily attributed to a stronger northern hemispheric summer insolation (Laskar et al., 2004) owing to the orbital configuration of higher eccentricity and obliquity (Berger, 1978), rather than higher greenhouse gas concentrations as projected for the future. The role of orbital forcing versus

40 greenhouse gases on temperature have been analysed in detail in Yin and Berger (2012). The LIG is associated with annual mean SSTs around 0.5°C higher than pre-industrial (PI) (Capron et al., 2014; Hoffman et al., 2017) and sea level 6-9 m higher than PI (Dutton et al., 2015; Rohling et al., 2019), although a recent study suggests the LIG sea level was 1.2-5.3 m higher than present day (Dyer et al., 2021). The area-weighted summer warming is estimated to be 1.1-1.9°C for the North Atlantic and 1.6-1.8°C for the SO compared to PI (Govin et al., 2012; Capron et al., 2014; Hoffman et al., 2017). Summers over land

areas are reconstructed to be 4-5°C warmer at high latitudes in the Northern Hemisphere (CAPE-Last Interglacial Project Members, 2006), 3-11°C over Greenland (Dahl-Jensen et al., 2013) and 2.2°C over Antarctica (Masson-Delmotte et al., 2013). Reconstructions of sea ice are mostly focused on the Arctic, and suggest ice-free conditions south of 78°N (Van Nieuwenhove et al., 2008, 2011, 2013; Kageyama et al., 2021). The Atlantic Meridional Overturning Circulation (AMOC) has been suggested to have weakened at the peak of the LIG (127ka) and strengthened afterwards (Galaasen et al., 2014), with strong evidence

pointing towards periods of reduced Antarctic Bottom Water (AABW) formation during the early part of the LIG, due to discharges from the Antarctic ice-sheet (Hayes et al., 2014; Rohling et al., 2019).

    A few studies have investigated the terrestrial carbon response to LIG conditions (Kleinen et al., 2015; Brovkin et al., 2016), but the marine carbon cycle response at the LIG, and particularly in the SO, has received little attention. A comprehensive study of the changes in the SO carbon cycle at the LIG compared to PI can enhance our understanding of the processes involved in

the SO carbon cycle, and their sensitivity to changes in boundary conditions. It can also help us, to a certain extent, to better quantify the impact of expected future physical and dynamical changes in a warming SO on the marine carbon cycle, bearing in mind that there was no additional anthropogenic carbon released during the LIG.

Here, we study the impact of a warmer climate, in particular at high latitudes, on the oceanic carbon cycle by analysing an equilibrium last interglacial simulation (lig127k, Otto-Bliesner et al. (2017)) performed with the ACCESS-ESM1.5 model (Ziehn et al., 2020; Yeung et al., 2021). The paper is structured as follows: In Section 2, components of the ACCESS-ESM1.5 model are described, followed by a framework for decomposing variables relevant for carbon cycle changes. The main results are presented in Section 3, discussing the changes in climate and the carbon cycle. These include differences in ocean properties, such as temperature, sea ice cover, stratification and circulation, and their effects on the carbon cycle, including air-sea fluxes, net carbon storage, and the efficiency of the biological pump. Finally in Section 4, we conclude and discuss the limitations of the current study including possible sources of uncertainty, and what inferences can be drawn for future global warming scenarios.

## 2 Methods

### 2.1 Model description and experimental design

An equilibrium last interglacial simulation (lig127k) is performed with the Australian Community Climate and Earth System Simulator Earth System Model, ACCESS-ESM1.5 (Ziehn et al., 2020), which includes interactive land and ocean carbon-cycles, and is Australia's submission to the Paleoclimate Modeling Intercomparison Project 4 (PMIP4) - Coupled Model Intercomparison Project (CMIP6) (Yeung et al., 2021). ACCESS-ESM1.5 differs from the previous version, ACCESS-ESM1 (Law et al., 2017), mostly in the land and ocean components. ACCESS-ESM1.5 is built upon the ACCESS1.4 physical model, and includes the UK Met Office Unified Model (UM) version 7.3 (Martin et al., 2010; Bellouin et al., 2011) as atmospheric component, which is directly coupled to the updated land Community Atmosphere Biosphere Land Exchange model (CABLE) version 2.4 (Kowalczyk et al., 2013), both with a horizontal resolution of $1.875° \times 1.25°$. UM has 38 vertical levels. The ocean model is the NOAA/GFDL Modular Ocean Model (MOM) version 5 (Griffies, 2014), which is coupled to the Los Alamos National Laboratory sea ice model (LANL CICE) version 4.1 (Hunke et al., 2010). MOM5's resolution is $1° \times 1°$ with 50 vertical levels. The coupler is the Ocean Atmosphere Sea Ice Soil - Model Coupling Toolkit (OASIS-MCT) (Craig et al., 2017). The current version of the land model CABLE also includes biogeochemistry (BGC) implemented using the CASA-CNP module (Wang et al., 2010). The ocean carbon cycle is simulated using the Whole Ocean Model of Biogeochemistry And Trophic-dynamics (WOMBAT) model (Oke et al., 2013).

WOMBAT is a nutrient–phytoplankton-zooplankton–detritus (NPZD) model, with one class of phytoplankton and one class of zooplankton. It includes DIC, alkalinity (ALK), phosphate ($PO_4$), oxygen ($O_2$) and iron. The biogeochemical tracers are coupled using the stoichiometric C (carbon):N (nitrogen):P (phosphorous):$O_2$ ratio of 106:16:1:172. The air-sea gas exchange is based on the square of the wind speed (Wanninkhof, 1992), and the seawater partial pressure of $CO_2$ ($pCO_2$) is calculated following the third phase of Ocean Carbon-Cycle Model Intercomparison Project (OCMIP) protocol using temperature, salinity, DIC, ALK and $PO_4$. WOMBAT simulates production in and export from the photic zone, and remineralization and dissolution at depth for both organic and inorganic ($CaCO_3$) particulate matter, with parameters adjusted for inorganic export to be ∼8% of organic export. The remineralization of organic matter is calculated based on depth and temperature dependent parameters,

while the dissolution of $CaCO_3$ uses a constant dissolution rate. ACCESS-ESM1.5 does not include a sediment component, and all particulates reaching the bottom ocean layer are instantly remineralized following the relevant remineralization rates. Other specific details of the model can be found in Oke et al. (2013), Law et al. (2017) and Ziehn et al. (2020).

A pre-industrial 1850 simulation (piControl) is run in accordance with the CMIP6 protocol (Eyring et al., 2016), with a constant atmospheric $CO_2$ forcing of 284.3 ppm, but using CMIP5 solar irradiance (1365.65 $Wm^{-2}$, as explained in Ziehn et al. (2020)). This simulation is run for 1000 years. Initialized from this PI control run, a LIG simulation (lig127k) is performed using orbital parameters following the PMIP4 protocol (Otto-Bliesner et al., 2017; Yeung et al., 2021), but with the solar constant adjusted to the CMIP5-PMIP3 value to be comparable to the piControl simulation (1365.65 $Wm^{-2}$). $CO_2$ concentration for the LIG is set at 275 ppm. Vegetation is kept constant at 1850 PI conditions. The lig127k experiment is run for 650 years. The run is at equilibrium over the SO, although there is a small drift in globally averaged DIC of ∼1 $\mu$mol/kg/100 years below 3 km. The analysis presented here is based on the average of the last 100 years of the lig127k simulation compared to the last 100 years of the piControl simulation. Further details on these experiments can be found in Otto-Bliesner et al. (2017), Ziehn et al. (2020), and Yeung et al. (2021).

WOMBAT includes two DIC tracers, one being forced by the prescribed atmospheric $CO_2$ concentration (PI: 284.3 ppm, LIG: 275 ppm), and the other forced by a constant atmospheric $CO_2$ concentration of 280 ppm. Unless mentioned otherwise, all of the analyses presented here are based on the tracers forced with $CO_2$ concentrations of 280 ppm, while the climate response is forced with the radiative forcing of 284.3 ppm and 275 ppm, respectively. This allows quantification of the effects of the LIG climate on the carbon cycle independently of the difference in background $CO_2$ concentrations.

### 2.1.1 Model Evaluation

After a 3000 years and 1000 years spin up of the physical and biogeochemical states, a 500 years piControl run was generated and used to assess the performances of the ACCESS-ESM1.5. The drifts in the piControl simulation are described in Ziehn et al. (2020) and are -8.5× $10^{-5}$°C century$^{-1}$ for SST, 5.3× $10^{-3}$°C century$^{-1}$ for global ocean temperatures, 7.6× $10^{-4}$ psu century$^{-1}$ for sea surface salinity (SSS), and 8.8× $10^{-4}$ psu century$^{-1}$ for global ocean salinity. The drift in total ocean productivity is -0.0163 PgC year$^{-1}$ per century, and that in carbon flux is -0.0048 PgC year$^{-1}$ per century. The net pre-industrial carbon flux is 0.02 PgC year$^{-1}$ and 0.08 PgC year$^{-1}$ for land and ocean respectively.

Ziehn et al. (2020) present results from the historical simulation performed with the ACCESS-ESM1.5, and evaluate the performance with respect to available observations. Here, we summarize the model performance pertaining to the SO. The model has a warm SST bias in the SO, possibly resulting from a too shallow and warm summer mixed layer, however, the climatological sea-ice extent in SO closely follows the observations (Fetterer et al., 2017). The ACCESS-ESM1.5 captures the surface patterns of nutrients relatively well, showing a 0.89 correlation with the surface phosphate distribution from World Ocean Atlas (Garcia et al., 2010b). The simulated primary productivity is higher over 40-45°S compared to observations (Behrenfeld and Falkowski, 1997), which leads to an increased uptake of $CO_2$ compared to the observed estimates (Takahashi et al., 2009). Compared to the GLODAP dataset (Key et al., 2004), phosphate and alkalinity concentrations are underestimated in the Southern Ocean by ∼0.4-0.8 mmol m$^{-3}$ and ∼50-100 mmol m$^{-3}$, respectively. In addition, the oxygen concentration

is $\sim$50-100 mmol m$^{-3}$ higher in the ACCESS-ESM1.5 across all depths (Garcia et al., 2010a). These biases could result from a strong ocean ventilation or a weak biological pump in the SO. Further details on model performance and evaluation can be found in Ziehn et al. (2020).

## 2.2 pCO$_2$ decomposition

Changes in surface pCO$_2$, which ultimately control the direction and magnitude of air-sea fluxes, can further be decomposed
into contributions from SST, SSS, DIC, and ALK (Sarmiento and Gruber, 2006) using the following equations:

$$\Delta pCO_2^{SST} = e^{(\gamma_{SST} \times \Delta SST)} \times pCO_2^{PI} - pCO_2^{PI} \tag{1}$$

where $\Delta pCO_2^{SST}$ represents the contribution of SST change to surface pCO$_2$ change, $\gamma_{SST}$=0.0423$^{\circ}$C$^{-1}$ is the sensitivity of pCO$_2$ changes to changes in SST (Sarmiento and Gruber, 2006). $\Delta SST$ is the difference in SST between LIG and PI, and $pCO_2^{PI}$ is the surface seawater pCO$_2$ from the piControl simulation.
And

$$\Delta pCO_2^{X} = \gamma_X \times \frac{\Delta X}{\overline{X}} \times pCO_2^{PI} \tag{2}$$

where $\Delta pCO_2^{X}$ represents the contribution of change in variable '$X$' (SSS, DIC, and ALK) to surface pCO$_2$ change, $\Delta X$ is the change in '$X$' between LIG and PI, and $\overline{X}$ is the mean value of '$X$' at PI, $\gamma_X$ is the sensitivity of pCO$_2$ changes to changes in variable '$X$' ($\gamma_{DIC}$ is known as the Revelle factor), with $\gamma_{SSS} = 1$,

$$140 \quad \gamma_{DIC} = \begin{cases} 15, & \text{if } lat \geq 60^{\circ} \\ 13, & lat < 60^{\circ} \end{cases}$$

and

$$\gamma_{ALK} = \begin{cases} -12, & \text{if } lat \geq 60^{\circ} \\ -11, & lat < 60^{\circ} \end{cases}$$

These values are based on previous estimates of the meridional profiles of DIC and ALK buffer factors (Sarmiento and Gruber, 2006; Smith and Marotzke, 2008; Egleston et al., 2010; Jiang et al., 2019), although uncertainties still remain regarding these
145 estimates. This decomposition helps shed light on the independent effects of different physical variables to the net surface pCO$_2$ change.

## 2.3 Carbon partitioning

To better quantify changes in the carbon cycle between the LIG and PI experiments, we split the total DIC into its remineralized (C$_{org}$), dissolved carbonate (C$_{CaCO_3}$), and preformed (C$_{pre}$) components using the equations listed below.
C$_{org}$ can be estimated from regenerated phosphate (PO$_4^{Reg}$) using the C:P stoichiometric ratio ($r_{C:P}$=106) (Ito and Follows, 2005):

$$\Delta C_{org} = r_{C:P} \times \Delta PO_4^{Reg} \tag{3}$$

$PO_4^{Reg}$ can be approximated using Apparent Oxygen Utilization (AOU) and the P:O$_2$ stoichiometric ratio ($r_{P:O_2}$=1/172) (Ito and Follows, 2005; Duteil et al., 2012):

$$\Delta PO_4^{Reg} = r_{P:O_2} \times \Delta AOU \tag{4}$$

AOU estimates the oxygen consumed during respiration and can be calculated as the difference of dissolved oxygen concentration (O$_2$) from the saturated concentration of oxygen (O$_2^{sat}$) (Weiss, 1970):

$$AOU = O_2^{sat}(T, S) - O_2 \tag{5}$$

This can then be used to infer the efficiency of the biological pump (BP$_{Eff}$) as per Ito and Follows (2005):

$$\overline{BP_{Eff}} = \overline{PO_4^{Reg}}/\overline{PO_4} \tag{6}$$

where $\overline{X}$ is the mean value of $X$.

The contribution to DIC from the carbonate pump is estimated by:

$$\Delta C_{CaCO_3} = 0.5 \times [\Delta ALK + r_{N:P} \times \Delta PO_4^{Reg}] \tag{7}$$

where the term '$r_{N:P} \times \Delta PO_4^{Reg}$' accounts for the reduction in ALK from production of nitrate (NO$_3^-$) (Sarmiento and Gruber, 2006), which is estimated using PO$_4^{Reg}$ and the N:P stoichiometric ratio ($r_{N:P}$=16). Finally, the preformed carbon concentration (C$_{pre}$) is obtained as:

$$\Delta C_{pre} = \Delta DIC - \Delta C_{org} - \Delta C_{CaCO_3} \tag{8}$$

## 3 Results

Changes in the climate system are first presented (Section 3.1), followed by their effects on air-sea gas exchange in the SO (Section 3.2). To understand these, we quantify the different contributors to changes in surface partial pressure of CO$_2$ next (Section 3.3), and finish by analysing deep ocean changes and the global oceanic carbon inventory (Section 3.4). Unless otherwise mentioned, the SO is defined as the ocean area south of 40°S.

### 3.1 Ocean dynamics and sea ice cover

As a result of the insolation anomalies and associated feedbacks, the global mean annual SST anomaly at the LIG compared to PI as simulated by the ACCESS-ESM1.5 equals to 0.17°C, with a pronounced warming at high latitudes and maximum positive SST anomalies over the North Atlantic (up to 4°C, Yeung et al. (2021)). Our simulated temperatures are in line with the range of PMIP4 lig127k simulations (Otto-Bliesner et al., 2021). A model-data comparison of the LIG climate state is presented in Yeung et al. (2021).

A mean 0.53°C warming is simulated over the SO south of 40°S (Fig. 1a). Warmer conditions are simulated everywhere south of 50°S apart from a ∼1°C cooling centered at 58°S in the South Atlantic, and up to a 1.5°C cooling in the subantarctic

eastern Pacific. The strongest warming is simulated over the southeast Atlantic and Indian Ocean sectors (up to 3°C). Regional SSTs up to 4°C higher are simulated around 60°S for both austral spring (Fig. A1b) and summer (not shown). The higher SSTs over the SO compared to PI are accompanied by a marked reduction in sea-ice extent over both austral summer and winter (Fig. 1a), peaking at 41% reduction in austral winter (Fig. A1a).

A 1.5° equatorward shift of the SH westerlies is simulated at the LIG (Fig. 1e), with a 10% weakening of the winds south of 50°S. This leads to ∼10% weaker upwelling south of 55°S and up to ∼20% stronger upwelling north of 55°S (Fig. 1f). Seasonally, the largest changes in upwelling are found for the winter and spring seasons (Fig. A1c and d). The equatorward shift of the westerlies reduces the northward Ekman transport south of 55°S leading to warming, while the higher Ekman transport north of 55°S induces a cooling around 50°S (Fig. 1a and b). The Antarctic Circumpolar Current is weaker, and the transport through Drake passage is reduced from PI by ∼55Sv (not shown).

## 3.2 Response of the air-sea gas exchange

These physical changes in the SO impact the carbon cycle and the air-sea $CO_2$ exchange (Fig. 2). It is worth reiterating here that we analyse changes in the carbon cycle using tracers for a constant atmospheric $CO_2$ concentration of 280 ppm for both simulations (Section 2.1), which enables us to solely analyse the impact of climatic and oceanic circulation changes on the carbon cycle. The $CO_2$ flux over the SO at PI shows a carbon uptake near the Antarctic coast, an outgassing band between 65-45°S, and another uptake zone further north (Fig. 2a and blue line in d). This $CO_2$ outgassing is due to the upwelling of DIC-rich deep waters (Fig. 1f). At the LIG, the upwelling region widens over the Atlantic and Indian ocean sectors, and narrows over the eastern Pacific sector (Fig. 2). There is a strong increase in outgassing in the Atlantic and western Indian, as well as at ∼67°S in the eastern Pacific sectors (Fig. 2c and red line in d). An increase in $CO_2$ uptake is simulated in the Amundsen, Bellinghausen, Weddell, Lazarev, Riiser-Larsen and Ross Seas, and the sub-antarctic east Pacific (Fig. 2c). The zonal mean $CO_2$ flux in Fig. 2d shows a small increase in uptake south of 62°S, possibly due to reduced mixing and reduced winter sea ice cover. Overall, there is a net increase in the mean SO outgassing by ∼150% at the LIG compared to PI (Fig. 2e). This increase in outgassing mostly occurs during the austral winter and spring seasons (Fig. A1e and f).

This increased outgassing over the SO is compensated by increased uptake over other ocean basins, especially the North Pacific subpolar gyre, the South Atlantic and the northwest Indian Ocean, while higher outgassing is simulated in the North Atlantic (Fig. A2). In this paper, we focus primarily on the SO. To better understand the SO changes, we decompose the oceanic $pCO_2$ changes into their different components.

## 3.3 Changes in surface carbon dioxide partial pressure

The air-sea gas exchange is primarily controlled by the seawater partial pressure of $CO_2$ ($pCO_2$). Using the equations presented in Section 2.2, Fig. 3 shows a decomposition of the changes in $pCO_2$ into their SST, SSS, DIC and ALK contributions. In line with the net increase in SO outgassing at the LIG (Fig. 2), the net $pCO_2$ over the SO is 1.2 $\mu$atm higher at the LIG compared to PI (black circle) (Fig. 3a). In agreement with the changes in air-sea $CO_2$ flux, surface $pCO_2$ is higher in a zonal band centered at ∼55°S, while it is lower in coastal regions (contours in Fig. 3b and black line in g). The contributions from individual

components add up to reflect the simulated differences reasonably well both in terms of magnitude (1.4 $\mu$atm, gray square in Fig. 3a) and spatial distribution (Fig. 3b), affirming the validity of the decomposition method. The overall slightly positive $pCO_2$ anomaly at the LIG compared to PI results from the competing effects of lower $CO_2$ solubility (red triangle, +5.65 $\mu$atm) and changes in surface DIC and ALK (blue diamond, -4.25 $\mu$atm).

The largest contributor to solubility is SST (brown triangle), while the largest contributor of the combined DIC and ALK effect is DIC (cyan diamond in Fig. 3a). Higher SSTs in the SO at the LIG lead to a 5.8 $\mu$atm $pCO_2$ increase (brown triangle in Fig. 3a). This SST induced increase is present over most of the SO, and is highest over the south Atlantic, Indian and west Pacific regions (Fig. 3c). Changes in SSS do not contribute significantly to the $pCO_2$ anomalies (magenta triangle in Fig. 3a and A3d). Changes in DIC cause the largest single contribution to the overall $pCO_2$ change with the net decrease in surface DIC leading to a -10.9 $\mu$atm $pCO_2$ change (cyan diamond in Fig. 3a). An exception are the higher DIC concentrations in the Atlantic Ocean around 55°S and in the eastern Pacific sector between 60°S-70°S (Fig. A1g,h, and A4d), leading to higher $pCO_2$ (Fig. 3e). This higher DIC is due to increased upwelling in these regions (Fig. 1b and A1c,d), resulting from the equatorward shift of the SH westerlies. The contributions based on changes in ALK and DIC have very similar patterns, albeit of opposite signs, with a small difference over the west Pacific around 50°S. The combined effect of ALK and DIC changes leads to a total net decrease in $pCO_2$ by 4.25 $\mu$atm due to the higher impact of changes in DIC (-10.9 $\mu$atm, cyan diamond) compared to ALK (+6.65 $\mu$atm, green diamond). Overall, south of 45°S, changes in SST lead to an overall increase in $pCO_2$ with the highest contributions over the Indian and West Pacific sectors of the SO, while changes in DIC lead to an overall decrease of $pCO_2$, with strongest reductions in the southern Atlantic, Indian and West Pacific sectors of the SO but an increase in the Amundsen Sea sector (Fig. 3).

The zonal mean patterns of these contributions are presented in Fig. 3g. Coastal regions around Antarctica, south of ~75°S, show that the lower $\Delta pCO_2$, both simulated (black line) and calculated (gray line), primarily arise from the combined DIC and ALK component (blue line) that is mainly attributed to the lower DIC (cyan dashed line) compensated by changes in ALK (green dashed line). Between 70°S and 60°S, $\Delta pCO_2$ anomalies are close to zero as the reduced $CO_2$ solubility due to higher SSTs (red line) is compensated by the combined DIC and ALK changes (blue line). Between 60°S and 45°S, the simulated surface ocean $\Delta pCO_2$ is positive (~2 $\mu$atm, black line) as the effect of higher SSTs dominates over the combined ALK and DIC components. North of ~45°S, the positive $\Delta pCO_2$ signal arises from the DIC+ALK component while solubility mitigates the anomaly (Fig. 3g).

Figure 3g shows that while solubility is mostly controlled by changes in SST, contributions from SSS changes are important south of 72°S. Similarly, DIC has the dominant control over the DIC+ALK component. The contributions from both the solubility and (DIC+ALK) components reach their maxima (solubility being positive and DIC+ALK being negative) around 62°S, which corresponds to the maximum divergence in wind-driven surface currents (Fig. 1f). To summarise, the higher overall $pCO_2$ (and hence lower air-sea $CO_2$ flux) at the LIG compared to PI can be attributed to higher SSTs south of 45°S, and the combined DIC-ALK component between 45°S and 35°S due to the equatorward shift of the upwelling regions (Fig. 3g). The SST patterns were already discussed in Section 3.1. DIC patterns (Fig. A4d) can result from changes in circulation, and the biological pump. These are investigated in the next section.

## 3.4 Deep ocean changes and carbon inventory

Figure 1c and d show that North Atlantic Deep Water (NADW) formation is ∼5 Sv higher in our LIG simulation compared to the PI simulation, leading to colder waters at depths of 1-2 km between 0-60°N, accompanied by higher SSTs and SSSs in the Labrador Sea (not shown). We also simulate a ∼4 Sv reduction in AABW formation (Fig. 1c and d). This reduction in AABW formation results in a warming of SO deep waters by up to 2°C compared to PI. Due to a northward shift and weakening of winds (Fig. 1e, f), the Antarctic Intermediate Water (AAIW) formation regions shift northward and the AAIW formation rate slows down (Downes et al., 2017). The changes in SST and upwelling (Fig. 1), alongside the reduced sea ice extent, lead to an increase in net primary production (NPP) and export production over the SO by ∼17% and ∼11% respectively (Fig. A4a and b).

These circulation changes impact the DIC distribution in the ocean (Fig. 4b). For instance, the reduced formation rate of AAIW (Fig. 1d) increases residence times and leads to lower dissolved oxygen (Fig. 4a), higher $PO_4$ (Fig. 4c), higher remineralized carbon (Fig. 4d) and higher DIC concentrations (Fig. 4b) at intermediate depths of the SO north of 55°S. Similarly, the increased accumulation of nutrients and DIC, as well as oxygen depletion in deep and abyssal waters (Fig. 4a, b and c), can be attributed to a weaker AABW formation rate (Fig. 1d), which results in a higher efficiency of the biological pump, as detailed below.

The ACCESS-ESM1.5 simulates a $\geq 50$ $\mu$mol/kg decrease in oxygen everywhere below 3 km at the LIG compared to PI. A decrease in oxygen can also be seen in the SO across all depths, including a northward reaching tongue attributable to AAIW at ∼1 km depth extending to the Equator (Fig. 4a). DIC anomalies follow the patterns of dissolved oxygen, albeit with the opposite sign. The simulated DIC concentrations are ∼50 $\mu$mol/kg higher across all depths in the SO and globally below 3.5 km depth (Fig. 4b).

DIC anomalies are decomposed into contributions from remineralized organic carbon, dissolved calcium carbonate, and preformed carbon components (Fig. 4d, f and h, Section 2.3). 60% of the DIC increase in the SO and abyssal ocean can be attributed to an increase in remineralised carbon (Fig. 4d), resulting from a 10% more efficient biological pump (Eq. 6, Section 2.3). This increase in remineralised carbon can be attributed to increased residence times, due to weaker bottom and intermediate water formation rates (Fig. 1 c and d). Weaker AABW and AAIW indeed lead to positive apparent oxygen utilization (AOU) and regenerated PO4 anomalies (Fig. 4e). In addition, a 2°C warming of bottom waters leads to a ∼6% increase in remineralization rates, thus further contributing to the higher remineralised carbon. Around 25% of the abyssal increase in DIC is attributed to the carbonate pump (Fig. 4f). This reflects changes in NPP, export production (Fig. A4) and water mass residence times, given that carbonate production is a constant percentage of total NPP in this model setup, and carbonate dissolution is constant and independent of water chemistry or temperature. Preformed DIC represents only ∼5% of the DIC changes (Fig. 4h). This increased sequestration of DIC in the deep ocean reduces the surface DIC concentration (Fig. A4 and A1g, h), thus contributing to a lowering of surface $pCO_2$ (Fig. 3e and g).

For the Northern Hemisphere, stronger NADW (Fig. 1c and d) results in decreased DIC concentrations in intermediate and deep waters of the North Atlantic by up to 75 $\mu$mol/kg (Fig. 4b). This negative DIC anomaly can be explained by a decrease

in both remineralised and preformed DIC. Stronger NADW subducts more DIC depleted surface waters into the deep ocean (Duteil et al., 2012), and reduces residence times, thus leading to a decrease in remineralized carbon. This is also associated with a ∼40 $\mu$mol/kg increase in dissolved oxygen content (Fig. 4a), and up to a 0.75 $\mu$mol/kg decrease in $PO_4$ concentration (Fig. 4c). The reduction in preformed carbon (Fig. 4h) is most likely due to a new NADW formation site in the Labrador Sea along with a slight northward shift of the deep water formation regions in the Norwegian and Greenland Seas.

## 4  Discussion and Conclusion

We analyse the SO marine carbon cycle response to warmer conditions as simulated in an equilibrium simulation of the LIG. The lig127k simulation performed with the ACCESS-ESM1.5 presented here displays an annual mean warming of 0.53°C at the surface of the SO compared to PI. This simulated southern high latitude warming (Fig. 1a) is in agreement with the multi-model mean of the PMIP4 lig127k simulations, although it is at the higher end of the spectrum (Otto-Bliesner et al., 2021). Seasonally, we simulate regional SSTs up to 4°C higher around 60°S for both the austral spring and summer, in line with both, proxy records (Capron et al., 2017; Hoffman et al., 2017) and the PMIP3 125ka experiment performed with the NORESM-1ME model (Kessler et al., 2018), while the PMIP4 lig127k multi-model mean for austral summer displays less than 0.5°C warming over the SO (Otto-Bliesner et al., 2021). The simulated SO sea ice concentration at the LIG for austral winter (Fig. 1 and A1) is also in good agreement with those inferred in previous studies (Holloway et al., 2016, 2017), even though the changes in Antarctic sea ice area of the ACCESS-ESM1.5 are the largest amongst all the CMIP6-PMIP4 models (Otto-Bliesner et al., 2021).

A 1.5° equatorward shift of the westerlies, resulting in a 10% weakening of the westerlies south of 50°S, is simulated at the LIG (Fig. 1e), leading to significant changes in SO upwelling (Fig. 1b and f). There is no clear consensus for position and strength of the SH westerlies during the LIG (Fogwill et al., 2014), although the higher obliquity might have led to a weakening of the westerlies compared to PI (Timmermann et al., 2014). These changes in winds lead to a northward shift of the AAIW formation regions. Our LIG simulation is also characterised by a weakening of AABW formation rates, which might be due to changes in surface density. Significant weakening of AABW during the LIG due to Antarctic meltwater discharge has previously been inferred from paleo-proxy records (Hayes et al., 2014; Rohling et al., 2019). A shift in westerlies might also contribute to a weakening of the AABW (Menviel et al., 2008; Huiskamp et al., 2016; Glasscock et al., 2020).

The reduced upwelling south of 55°S and weaker AABW transport simulated here lead to an increased sequestration of DIC in the deep ocean, through an increase in the efficiency of the biological pump (Fig. 4). This reduces the surface DIC concentrations, leading to a net reduction in outgassing over the SO, as has been previously hypothesised (Toggweiler et al., 2006). However, while the combined DIC and ALK contributions would lead to a lower $pCO_2$ at the surface of the SO, this change is overcompensated by the warmer conditions (Fig. 3). Reduced solubility due to higher SSTs leads to an increase in outgassing over most of the SO, while the reduced sea-ice cover does not seem to significantly impact the $CO_2$ fluxes. Assessing the impact of sea-ice changes on $CO_2$ fluxes (Appendix A), we find that reduced sea-ice concentration at the LIG in the Weddell and Ross Seas leads to a 5% increase in $CO_2$ uptake in autumn and winter (Fig. A5). This results in a net outgassing

of $CO_2$ over the SO, with a $\sim$150% increase at the LIG compared to PI (Fig. 2), and the largest increase over the austral winter and spring seasons (Fig. A1). The simulated NPP and export production are $\sim$17% and $\sim$11% higher, respectively, over the SO at the LIG compared to PI (Fig. A4), providing a negative feedback to the higher outgassing. Although, the simulated nutrient and alkalinity concentrations are underestimated over the SO, the vertical gradients of these are captured reasonably well (Ziehn et al., 2020). As such, these biases should not significantly impact our results. All our analysis is based on a constant atmospheric $CO_2$ concentration of 280 ppm to allow quantification of the effects of the LIG climate on the carbon cycle independently of the background $CO_2$ concentration. However, this constant atmospheric $CO_2$ concentration neglects feedbacks related to $CO_2$ uptake and outgassing. Nevertheless, the lower $CO_2$ at LIG (275 ppm) compared to PI (284.3 ppm) would suggest the LIG SO to be an even greater $CO_2$ source to the atmosphere, implying a stronger sink somewhere else in the ocean or on land (Brovkin et al., 2016).

Numerical studies have unequivocally shown the impact of changes in the magnitude of the westerlies on the carbon cycle, with stronger westerlies leading to increased upwelling and SO outgassing, and vice-versa (e.g., Menviel et al., 2008; d'Orgeville et al., 2010; Lauderdale et al., 2013; Huiskamp et al., 2016; Lauderdale et al., 2017; Menviel et al., 2018; Gottschalk et al., 2019), however, the impact of changes in the latitudinal position of the westerlies on the carbon cycle is less certain (e.g., d'Orgeville et al., 2010; Völker and Köhler, 2013; Lauderdale et al., 2013; Huiskamp et al., 2016). Given the simulated increase in DIC content in the deeper SO and the reduced DIC at the surface of the SO, our simulations suggest that the simulated equatorward shift of the westerlies reduces the SO $CO_2$ outgassing.

SH westerlies are projected to strengthen and shift poleward over the coming century (Collins et al., 2013; Zheng et al., 2013; Goyal et al., 2021), contrary to the LIG simulations presented here. Also, while we find a more efficient biological pump at the LIG, Boyd (2015) suggested the biological pump to be less efficient under a future warming scenario, although recent studies have suggested a possible increase in surface productivity in the future (Kwiatkowski et al., 2020). Thus, changes in the carbon cycle simulated at the LIG may not serve as a good analogue for potential future changes. Nevertheless, the simulated enhanced SO $CO_2$ outgassing, despite a slight equatorward shift of the westerlies, support a weaker capability of the Southern Ocean to take up anthropogenic $CO_2$ over the coming century.

*Code and data availability.* Outputs of the physical variables from the lig127k simulation are archived on the CMIP6 ESGF website at https://doi.org/10.22033/ ESGF/CMIP6.13703 (Yeung et al., 2019). Outputs of the carbon cycle model have already been CMORised, and will be available on the ESGF website by the end of August 2021.

*Author contributions.* DC performed all of the analysis and writing of the results. LM and KJM provided support for interpretation and writing of the results. NKHY performed the lig127k simulation. MC and TZ contributed to the model setup and troubleshooting. All authors contributed towards the final manuscript.

*Competing interests.* The authors declare no competing interests

*Acknowledgements.* The simulations were conducted and analyzed at the National Computing Infrastructure (NCI) National Facility at the Australian National University, through awards under the Merit Allocation Scheme, the Intersect allocation scheme, and the UNSW HPC at NCI scheme. This research has been supported by the Australian Research Council (grant nos. DP180100048 and FT180100606).

**Appendix A: Effect of sea ice on air-sea gas exchange**

To estimate the effect of sea ice changes at the LIG on the air-sea gas exchange, we use a modified version of the equation of $CO_2$ flux from Wanninkhof (2014):

$$F_{CO_2} = 7.7 \times 10^{-4} \times |U^2| \times \Delta pCO_2 \times (1 - Sc) \tag{A1}$$

where $F_{CO_2}$ is the $CO_2$ flux ($molCm^{-2}y-1$), U is the average wind speed ($ms^{-1}$), $\Delta pCO_2$ is the difference in partial pressure
of $CO_2$ between ocean surface and atmosphere ($\mu$atm) and Sc is the sea ice concentration. To investigate the effect of LIG sea ice changes on the LIG $CO_2$ flux, we estimate $F_{CO_2}$ using both LIG Sc as well as PI Sc, while keeping all the other variables at LIG values. The resulting patterns are presented in Fig. A5. Figure A5 shows that the reduced sea ice during LIG compared to PI leads to less than 5% increase in carbon sink over the Ross Sea region all year round and Weddell Sea region over autumn and winter, and around 2% increase in outgassing in the Lazarev and Weddell Seas.

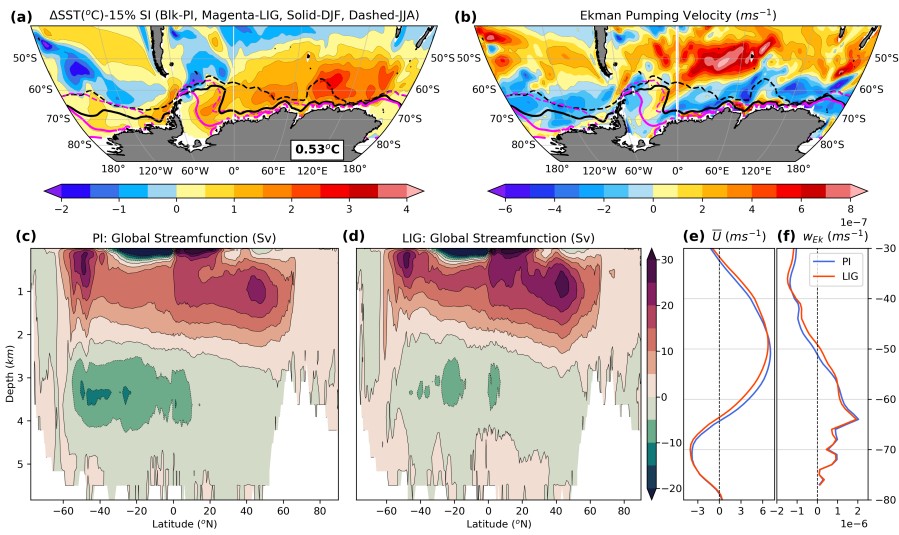

**Figure 1.** Annual mean (a) SST anomalies (°C) between LIG and PI overlaid with 15% sea ice concentration (black for PI, magenta for LIG, solid lines for DJF, dashed lines for JJA). Value in the box shows the SO (40-90°S) mean $\Delta$SST. (b) Anomalies of the Ekman pumping velocities between LIG and PI ($ms^{-1}$) overlaid with sea ice concentration (same as (a)). (c) Global mean meridional streamfunction ($Sv$) for PI and (d) LIG. Positive values indicate clockwise water mass transport and negative values counterclockwise transport. Zonal mean (e) winds ($ms^{-1}$), and (f) Ekman pumping velocities ($ms^{-1}$) over SO for PI (blue) and LIG (red).

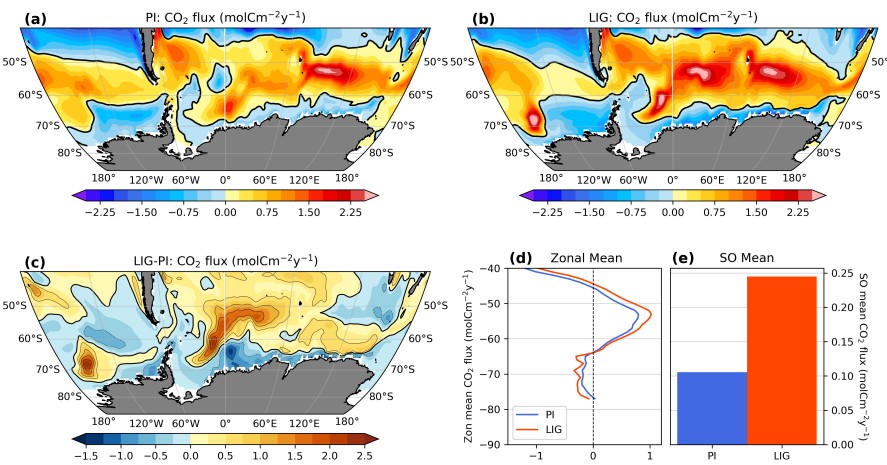

**Figure 2.** Annual mean air-sea $CO_2$ flux ($molCm^{-2}y^{-1}$) for (a) PI, (b) LIG, and (c) LIG-PI. Red colors indicate outgassing of $CO_2$ from the ocean, and blue uptake by the ocean. Thick black lines show the zero line contour. (d) Zonal mean $CO_2$ flux ($molCm^{-2}$ $y^{-1}$) over the SO for PI (blue) and LIG (red). (e) Mean $CO_2$ flux over the SO ($molCm^{-2}y^{-1}$) for PI (blue) and LIG (red).

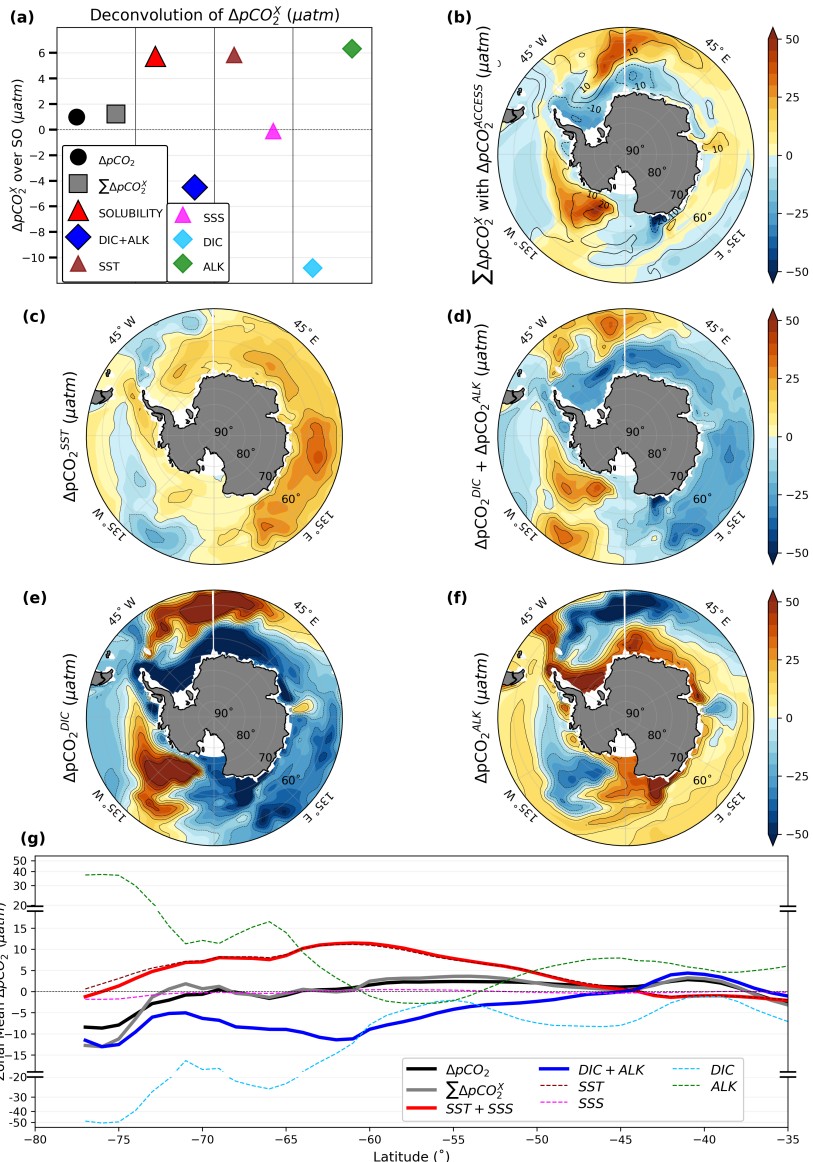

**Figure 3.** Attribution of changes in annual mean surface partial pressure of $CO_2$ (pCO₂) over SO ($\mu$atm). (a) Summary of decomposition of pCO₂ over SO. Simulated difference in pCO₂ (black circle) and sum of contributions from all components (gray square, Section 2.2). pCO₂ change from solubility (SST+SSS, red triangle), and sum of DIC and ALK components (blue diamond). Individual contributions from SST (brown triangle), SSS (magenta triangle), DIC (cyan diamond), and ALK (green diamond). (b) Map of the sum of pCO₂ contributions from all components (corresponding to gray square in (a)) in shading, overlaid with the pCO₂ change simulated by the model as contours (corresponding to black circle in (a)). Maps of individual pCO₂ contributions from (c) SST (corresponding to the brown triangle in (a)), (d) DIC and ALK (corresponding to the blue diamond in (a)), (e) DIC (corresponding to the cyan diamond in (a)), and (f) ALK (corresponding to the green diamond in (a)). (g) Zonal mean contributions to pCO₂ over the SO ($\mu$atm) for simulated $\Delta$pCO₂ (black), sum of contributions from all components (gray), solubility (red), sum of DIC and ALK components (blue), SST (dashed brown), SSS (dashed magenta), DIC (dashed cyan), and ALK changes (dashed green). Note the non-linear scale on the top and bottom thirds of (g).

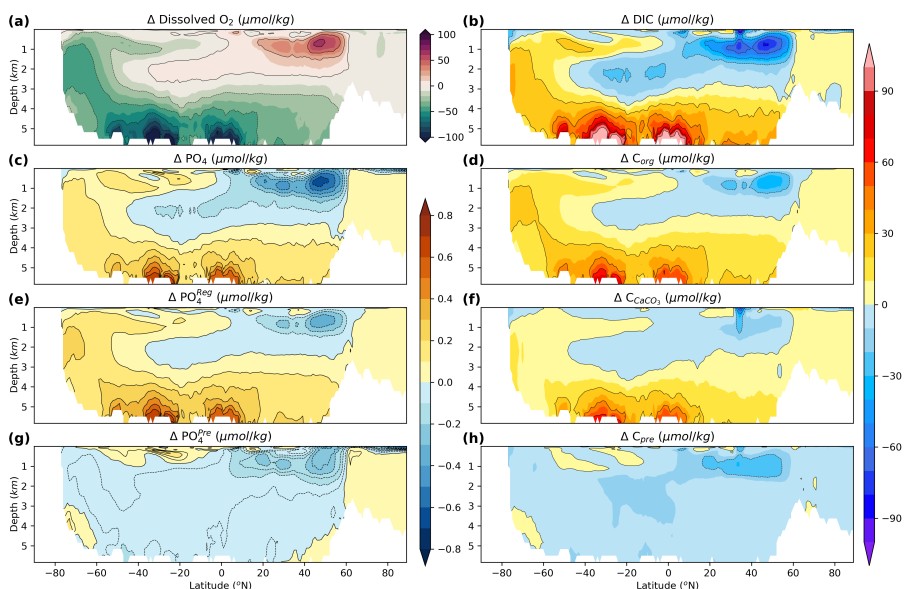

**Figure 4.** Global marine carbon budget and decomposition. Global zonal and annual mean anomalies of (a) dissolved oxygen concentration ($\mu$mol/kg), (c) phosphate concentration ($\mu$mol/kg), (e) regenerated phosphate concentration ($\mu$mol/kg), (g) preformed phosphate concentration ($\mu$mol/kg), (b) total DIC ($\mu$mol/kg), (d) remineralized carbon ($\mu$mol/kg), (f) dissolved carbonate ($\mu$mol/kg), and (h) preformed DIC ($\mu$mol/kg). Note that the phosphate components (c, e, and g), and the DIC components (b, d, f and h) each share common colorbars.

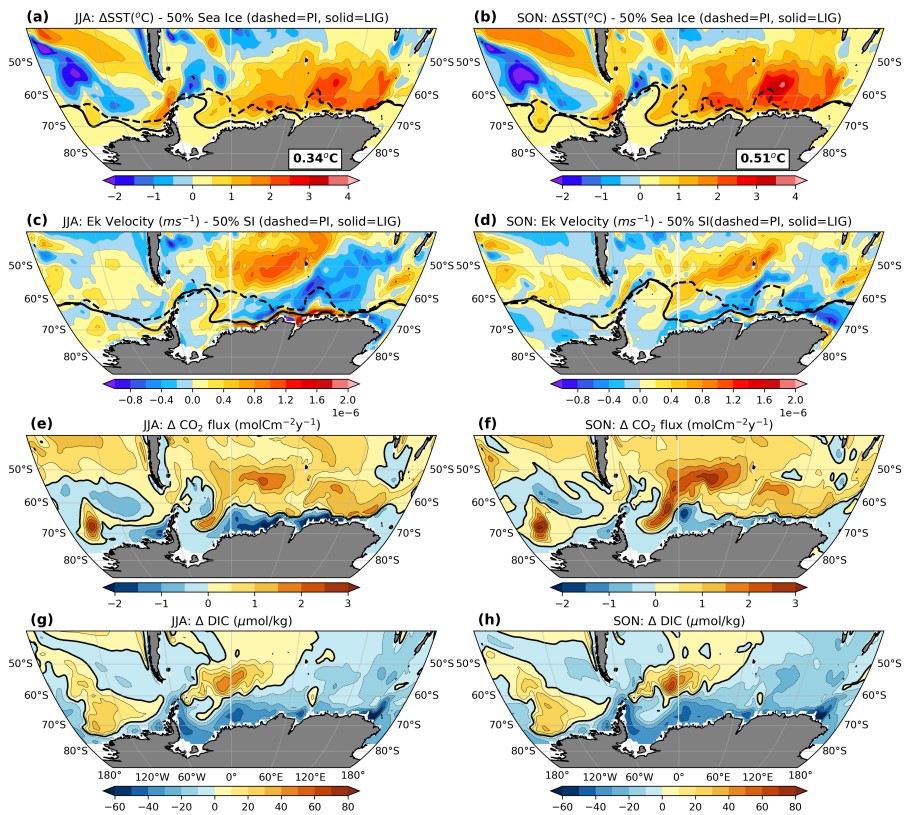

**Figure A1.** Seasonal anomalies for austral winter, JJA (1st column, a, c, e, g), and austral spring, SON (2nd column, b, d, f, h) of (a,b) SST ($^\circ$C) overlaid with 50% sea ice concentration (dashed for PI and solid for LIG), (c,d) Ekman pumping velocities ($ms^{-1}$) overlaid with 50% sea ice concentration (dashed for PI and solid for LIG), (e,f) sea-air $CO_2$ flux (mol C m$^{-2}$y$^{-1}$, positive indicating outgassing from and negative uptake by the ocean), and (g,h) surface DIC concentrations ($\mu$mol/kg).

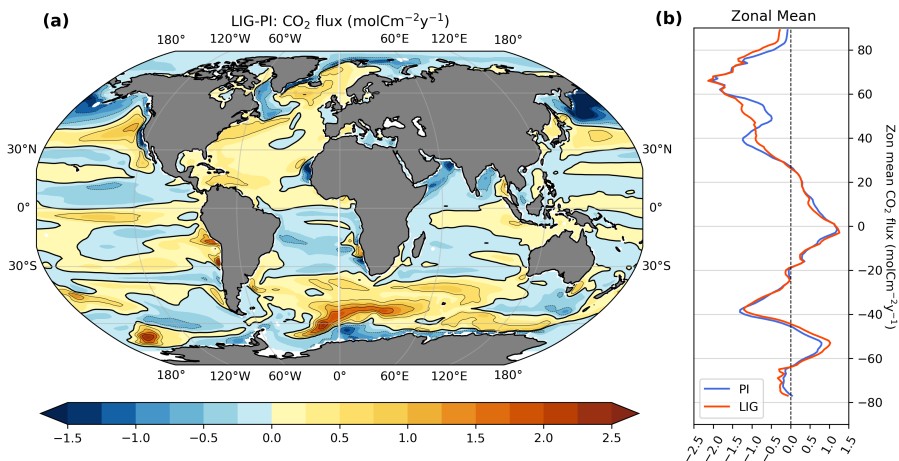

**Figure A2.** (a) Annual mean air-sea $CO_2$ flux $(molCm^{-2}y^{-1})$ for LIG-PI. Red colors indicate outgassing of $CO_2$ from the ocean, and blue uptake by the ocean. Thick black lines show the zero line contour. (b) Zonal mean $CO_2$ flux $(molCm^{-2} y^{-1})$ for PI (blue) and LIG (red).

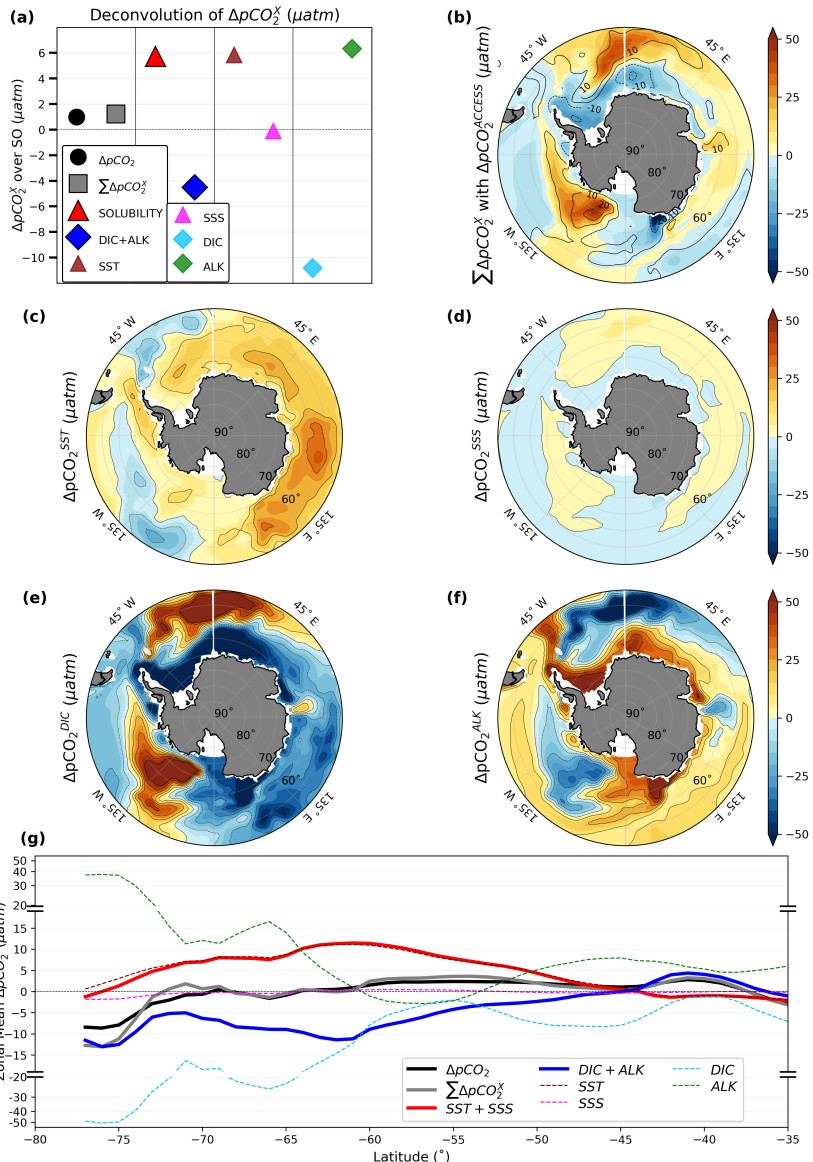

**Figure A3.** Attribution of changes in annual mean surface partial pressure of $CO_2$ (p$CO_2$) over SO ($\mu atm$). (a) Summary of decomposition of p$CO_2$ over SO. Simulated difference in p$CO_2$ (black circle) and sum of contributions from all components (gray square, Section 2.2). p$CO_2$ change from solubility (SST+SSS, red triangle), and sum of DIC and ALK components (blue diamond). Individual contributions from SST (brown triangle), SSS (magenta triangle), DIC (cyan diamond), and ALK (green diamond). (b) Map of the sum of p$CO_2$ contributions from all components (corresponding to gray square in (a)) in shading, overlaid with the p$CO_2$ change simulated by the model as contours (corresponding to black circle in (a)). Maps of individual p$CO_2$ contributions from (c) SST (corresponding to the brown triangle in (a)), (d) SSS (corresponding to the magenta triangle in (a)), (e) DIC (corresponding to the cyan diamond in (a)), and (f) ALK (corresponding to the green diamond in (a)). (g) Zonal mean contributions to p$CO_2$ over the SO ($\mu atm$) for simulated $\Delta$p$CO_2$ (black), sum of contributions from all components (gray), solubility (red), sum of DIC and ALK components (blue), SST (dashed brown), SSS (dashed magenta), DIC (dashed cyan), and ALK changes (dashed green). Note the non-linear scale on the top and bottom thirds of (g).

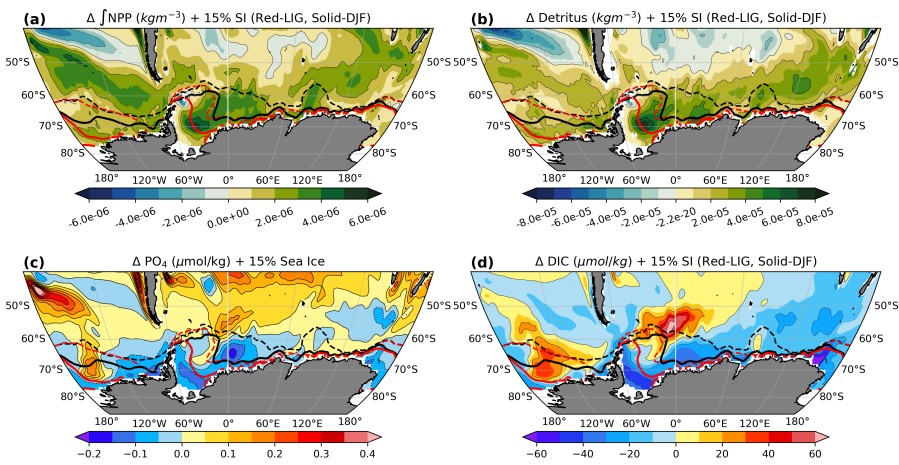

**Figure A4.** Changes in annual mean productivity and related variables. (a) Anomalies of depth integrated annual mean NPP (as cholorphyll in seawater, kgm$^{-3}$) overlaid with 15% sea ice concentration (black for PI, red for LIG, solid lines for DJF, dashed lines for JJA). (b) Anomalies of annual mean detrital concentration at 200m depth (kgm$^{-3}$) overlaid with sea ice (same as (a)). (c) Anomalies of annual mean surface phosphate concentrations ($\mu$mol/kg) overlaid with sea ice (same as (a)). (d) Anomalies of annual mean surface DIC concentrations ($\mu$mol/kg) overlaid with sea ice (same as (a)).

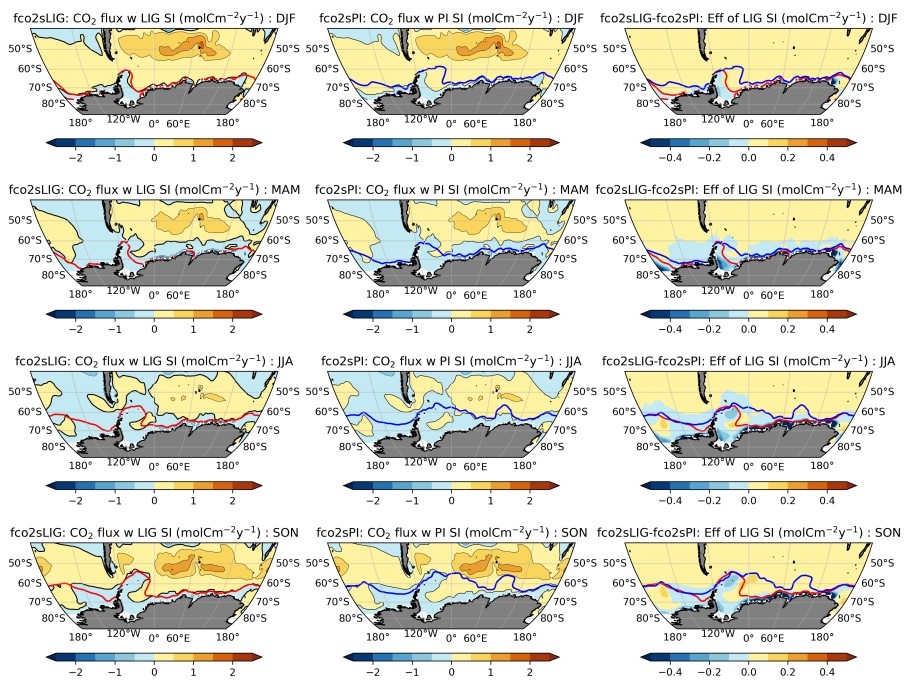

**Figure A5.** Calculated seasonal $CO_2$ fluxes using LIG $\Delta pCO_2$, winds and (left) LIG sea-ice concentration, (middle) PI sea ice concentrations; (right) differences in $CO_2$ flux for calculations using the LIG sea ice concentrations and compared to PI sea ice concentrations. Notice the reduced color scale in the third column. Calculations are based on Wanninkhof (1992) and Wanninkhof (2014). The red and blue contours indicate 15% sea ice concentrations for LIG and PI respectively.

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
