# Peer review of "Marine carbon cycle response to a warmer Southern Ocean: the case of the Last Interglacial"

_Climate of the Past, 2021_

## Author Comment (AC1)

We thank the Reviewer for their constructive comments, which have helped improve the quality of the manuscript. Please find detailed responses to the comments raised below along with excerpts from the revised manuscript given in boxes.

We have worked in particular on the introduction and discussion sections to emphasize the differences between changes expected in the future and the changes we analyze here based on our Last Interglacial simulation.

**Reviewer 1:**

**General comment:**

*This paper analyses the Southern Ocean carbon cycle from simulations of the Australian Community Climate and Earth System Simulator Earth System Model, ACCESS-ESM1.5, which has been applied for the PMIP4 scenario lig127k on the Last Interglacial (LIG). The argument is made that since this was the warmest interglacial of the last 1 million year it might serve as an analog for changes which might be expected in the future due to anthropogenically caused glabal warming.*

*The methods and results are well written, and the figure are very informative. I suggest some improvement in the introduction during the framing of the research question (see details below).*

*However, my major point is that I have some difficulties with this suggested analogy of the Last Interglacial with future warming. The analysis presented here shows that during the simulated LIG the westerly winds have been shifted equatorwards resulting in weakenend winds south of 50°S. This process is then responsible for different upwelling pattern and is important for quite a bit of changes in the Southern Ocean carbon cycle. For the future warming, it is now anticipated that westerly winds will shift polewards and get stronger, thus the opposite of what has been found for the LIG. I therefore strongly suggest to reframe the article in a way that its interpretation is largely restricted to the LIG. This reframing probably includes a change in the title. I would even go so far in pointing out in the dicsussion, that due to these shifts in wind pattern found here the LIG is no good analog for what to expect from future warming for the Southern Ocean carbon cycle. The reason for these differences in winds patterns are suggested to be due the the changes in orbital parameters (which seems to make sense), and this shows that past analogies for the future are often problematic.*

We agree with the Reviewer's comment and have now changed the manuscript accordingly. Changes in the westerlies as simulated in our LIG experiment are indeed opposite to the observed changes over the past 20 years and to the projected changes over the coming century. This was mentioned on p1 L. 21, p2 L. 28 and p10 L. 297. We are now further clarifying the differences between our LIG simulation and future projections, as presented below. However, since simulated Southern Ocean SSTs are higher than PI and sea-ice cover is reduced by up to 41%, we do believe that our LIG simulation allows us to study the carbon cycle response to generally warmer conditions. We therefore decided not to change the title, which is very general and does not refer to future climate change.

    i.    Abstract:

> The projected strengthening and poleward shift of the SH westerlies coupled to warmer conditions at the surface of the SO should thus weaken the capacity of the SO to absorb anthropogenic $CO_2$ over the coming century.

ii. Introduction in p1-2 L. 19-23:

> The SO $CO_2$ uptake weakened during the 1990s due to a strengthening and poleward shift of the Southern Hemisphere (SH) mid-latitude westerlies (Lovenduski et al., 2007; Le Quéré et al., 2007; Zickfeld et al., 2007, Gruber et al., 2019a, b), but strengthened in the 2000s due to cooling over the Pacific and increased stratification over the Atlantic and Indian sectors of the SO (Landschuetzer et al., 2016; Gruber et al., 2019b).

iii. Introduction in p2, L. 28-29:

> At the same time, SH westerlies are projected to strengthen and shift poleward over the coming century (Collins et al., 2013; Zheng et al., 2013; Goyal et al., 2021).

iv. We have removed the comparison with future simulations in P9, L263-264 in the Discussion.

v. We have also completely reworded the concluding statements to explicitly clarify that the LIG is not a good analogue for future changes. The current concluding paragraph in Discussion in P10, L. 302-306 now reads:

> SH westerlies are projected to strengthen and shift poleward over the coming century (Collins et al., 2013; Zheng et al., 2013; Goyal et al., 2021), contrary to the LIG simulations presented here. Thus, changes in the carbon cycle simulated at the LIG may not serve as good analogues for potential future changes. Nevertheless, the simulated enhanced SO $CO_2$ outgassing despite a slight equatorward shift of the westerlies support a weaker capability of the Southern Ocean to take up anthropogenic $CO_2$.

**Detailed comments:**

*lines 18-19: „of which 40% has been attributed to the SO" It is not clear on what this 40% is related to since before it is said "Both land and ocean act as sink". Does it refer only to the ocean part? And you probably mean that land and ocean EACH absorbs 25% of the anthropogenic emissions. You might furthermore consider citing the most recent version of the global carbon project here, thus „Friedlingstein et al 2020" (instead of 2019) https://doi.org/10.5194/essd-12-3269-2020â*

A. We have now clarified this. The text now reads:

> Both land and ocean presently act as sinks of anthropogenic carbon, each absorbing about 25% of anthropogenic emissions (Friedlingstein et al., 2020), with 40% of the ocean sink being attributed to the SO (Caldeira and Duffy, 2000; DeVries, 2014).

*line 35: „The Last Interglacial (LIG, 129-115 thousand years ago, ka) was the warmest interglacial of the last million years". This statement is problematic, since the paper PAGES2016 cited here analyse only the last 800 kyr.*

A. We have corrected this in the text, which now reads:

> The Last Interglacial (LIG, 129-115 thousand years ago, ka) was the warmest interglacial of the last 800 thousand years (Masson-Delmotte et al., 2013; PAGES, 2016).

*line 36ff: „The warmer climate at the LIG is primarily attributed to a stronger northern hemispheric summer insolation (Laskar et al., 2004) owing to the orbital configuration of higher eccentricity and obliquity (Berger, 1978), rather than higher greenhouse gas concentrations as projected for the future". The role of orbital forcing vs greenhouse gases on temperature have been analysed in detail in Yin and Berger 2012 DOI 10.1007/s00382-011-1013-5.*

A. We now added this in the text:

> The warmer climate at the LIG is primarily attributed to a stronger northern hemispheric summer insolation (Laskar et al., 2004) owing to the orbital configuration of higher eccentricity and obliquity (Berger, 1978), rather than higher greenhouse gas concentrations as projected for the future. The role of orbital forcing versus greenhouse gases on temperature have been analysed in detail in Yin and Berger (2012).

*line 38: „The LIG is associated with sea levels 6-9 m higher than pre-industrial (PI) (Dutton et al., 2015)". This knowledgee on LIG sea level has recently been revised downward, please reframe according to Dyer etal (2021) https://doi.org/10.1073/pnas.2026839118.*

A. We have now amended the text and added a reference to (Dyer et al., 2021):

> The LIG is associated with annual mean SSTs around 0.5˚C higher than pre-industrial (PI) (Capron et al., 2014; Hoffman et al., 2017) and sea level 6-9 m higher than PI (Dutton et al., 2015, Rohling et al., 2019), although a recent study suggests the LIG sea level was 1.2 to 5.3 m higher than present-day (Dyer et al., 2021).

*lines 91-94: Two different DIC tracers. I cannot remember that one of the tracers (the one not being constant at 280 ppm) is ever mentioned again. If so, it can be deleted here. You should also mention here, that since atmospheric CO2 is prescribed this approach misses the feedbacks which are related to CO2 in/outgassing. Also, absolute CO2 fluxes are biased since*

*the C cycle is simplified by this fixed CO2, which is acceptable for these interglacial conditions, but nevertheless might introduce a bias.*

A. We think that it is best to mention the two different tracers to avoid confusion, as in most models the $CO_2$ value used for the radiative forcing would also be used to force the marine carbon cycle. However, the Reviewer raises valid concerns regarding missing feedbacks and biases, and we have now addressed these concerns in the discussion section, and have added the following:

> All our analysis is based on a constant atmospheric $CO_2$ concentration of 280 ppm to allow quantification of the effects of the LIG climate on the carbon cycle independently of the background $CO_2$ concentration. However, this constant atmospheric $CO_2$ concentration neglects feedbacks related to $CO_2$ uptake and outgassing. Nevertheless, the lower $CO_2$ at LIG (275 ppm) compared to PI (284.3 ppm) would suggest the LIG SO to be an even greater $CO_2$ source to the atmosphere, implying a stronger sink somewhere else in the ocean or on land (Brovkin et al., 2016).

*line 110: $\gamma_{SST}$ = 0.0423°C$^{-1}$ is called the Revelle factor. I am completely lost here. For me, the Revelle factor R is the relative change in atm CO2 over the relative change in DIC (unitless) R = Δ(CO2)/CO2 / Δ(DIC)/DIC, eg. Egleston et al (2010) doi:10.1029/2008GB003407, while you here describe some temperature-dependency. Please revise, or explain.*

A. The reviewer correctly points out that the Revelle factor for DIC is unitless and represented as $R = \Delta(CO_2)/CO_2/ \Delta(DIC)/DIC$ (Equation 2). However, the temperature sensitivity of $CO_2$ is slightly more complicated. This is because the equilibrium constant for solubility itself has a temperature dependence. This temperature dependence of solubility leads to a logarithmic relationship between temperature and $CO_2$ (Equation 8.3.4 in Sarmiento and Gruber, 2006), and has been verified experimentally (Takahashi et al., 1993):

$$\frac{1}{pCO_2}\frac{\partial pCO_2}{\partial T} = \frac{\partial \ln pCO_2}{\partial T} = \gamma_{SST} \approx 0.0423^o C^{-1}$$

Whereas for DIC, the sensitivity is unitless and referred to as the 'Revelle factor' (Equation 8.3.14 in Sarmiento and Gruber, 2006):

$$\frac{DIC}{pCO_2}\frac{\partial pCO_2}{\partial DIC} = \frac{\partial \ln pCO_2}{\partial \ln DIC} = \gamma_{DIC}$$

We have now clarified the temperature 'sensitivity' in the methods section 2.2.

> $\gamma$ =0.0423°C$^{-1}$ is the sensitivity of $pCO_2$ to changes in SST (Sarmiento and Gruber, 2006). This SST sensitivity results from the equilibrium constant of solubility being temperature-dependent.

*line 19: weaker and stronger upwelling: by how much stronger or weaker?*

A. We have now updated the text to clarify the changes in upwelling in section 3.1, which now reads:

This leads to ~10% weaker upwelling south of 55˚S and up to ~20% stronger upwelling north of 55˚S.

*Fig 3g: xaxis title is missing*

A. This was fixed, thank you.

**Reference:**
1. Dyer, B., Austermann, J., D'Andrea, W. J., Creel, R. C., Sandstrom, M. R., Cashman, M., Rovere, A., and Raymo, M. E.: Sea-level trends across The Bahamas constrain peak last interglacial ice melt, Proc. Natl. Acad. Sci., 118, 2021.
2. Sarmiento, J. L. and Gruber, N.: Ocean biogeochemical dynamics, Princeton University Press, 2006.
3. Takahashi, T., Olafsson, J., Goddard, J. G., Chipman, D. W., and Sutherland, S. C.: Seasonal variation of $CO_2$ and nutrients in the high-latitude surface oceans: A comparative study, Global Biogeochem. Cycles, 7, 843–878, 1993.

---

## Author Comment (AC2)

We thank the Reviewer for their constructive comments, which have helped improve the quality of the manuscript. Please find detailed responses to the comments raised below along with excerpts from the revised manuscript given in boxes.

We have worked in particular on emphasizing the differences between changes expected in the future and the changes we analyse here based on our Last Interglacial simulation. We have also added relevant information on the model evaluation and on how the model biases can impact our results. Additionally, we have performed offline calculations to better ascertain the impact of sea ice changes on the air-sea gas exchange.

**Reviewer 2:**

**General comment:**

*The authors present the ocean carbon cycle response to a warmer climate, as simulated by an Earth System Model with the last interglacial (LIG) radiative forcing. The authors compared the Southern Ocean air-sea CO2 exchanges between the LIG and the pre-industrial (PI), and attributed the difference to changes in SST, SSS, DIC, and alkalinity using a decomposition method. A major finding includes a greater CO2 outgassing during the LIG compared to the PI, caused by warmer SST overwhelming the effects of surface DIC decreases. As the authors noted, exploring the Southern Ocean carbon cycle sensitivity to a warmer climate is an important research topic. The analyses/interpretations of the model results are also convincing. However, I have a few comments that might help improve this study.*

1. *Although this study focuses on the potential changes in the Southern Ocean carbon cycle during the LIG, there are no discussions of how the simulated changes are compared with other observation or model based studies. The authors discussed it for some physical variables such as SST and sea ice extent, but no discussions regarding biogeochemical properties (which are the focus of this work).*

And

2. *There are no validations of the model SST, SSS, DIC, and alkalinity against observations. Climatological mean DIC and alkalinity data are available through the GLODAP project where the authors can download observation-based estimates for the preindustrial period. The validation could be important because the simulated depth gradients of DIC and alkalinity can affect their redistributions during the LIG, which in turn control ocean surface pCO2 changes through their surface changes. Of course, the model would not be perfect, but potential biases due to the model deficiencies need to be discussed.*

A. We agree with the reviewer that validation of the model and inherent biases are important to better understand the changes simulated by the model. The model performances for the pre-industrial and historical simulations are described in detail in Ziehn et al. (2020). In addition, we have now added a section summarizing the model performances relevant to our study:

**2.1.1 Model Evaluation**

After a 3000 years and 1000 years spin up of the physical and biogeochemical states, a 500 years piControl run was generated and used to assess the performances of the ACCESS-ESM1.5. The drifts in the piControl simulation are described in Ziehn et al., (2020) and are -8.5×10$^{-5}$°C century$^{-1}$ for SST, 5.3×10$^{-3}$°C century$^{-1}$ for global ocean temperatures, 7.6×10$^{-4}$ psu century$^{-1}$ for sea surface salinity (SSS), and 8.8×10$^{-4}$ psu century$^{-1}$ for global ocean salinity. The drift in total ocean productivity is -0.0163 PgC year$^{-1}$ per century, and that in carbon flux is -0.0048 PgC year$^{-1}$ per century. The net pre-industrial carbon flux is 0.02 PgC year$^{-1}$ and 0.08 PgC year$^{-1}$ for land and ocean respectively.

Ziehn et al. (2020) present results from the historical simulation performed with the ACCESS-ESM1.5, and evaluate the performance with respect to available observations. Here, we summarize the model performance pertaining to the SO. The model has a warm SST bias in the SO, possibly resulting from a too shallow and warm summer mixed layer, however, the climatological sea-ice extent in SO closely follows the observations (Fetterer et al., 2017). The ACCESS-ESM1.5 captures the surface patterns of nutrients relatively well, showing a 0.89 correlation with the surface phosphate distribution from World Ocean Atlas (Garcia et al., 2010b). The simulated primary productivity is higher over 40-45°S compared to observations (Behrenfeld and Falkowski, 1997), which leads to an increased uptake of $CO_2$ compared to the observed estimates (Takahashi et al., 2009). Compared to the GLODAP dataset (Key et al., 2004), phosphate and alkalinity concentrations are underestimated in the Southern Ocean by ~0.4-0.8 mmol m$^{-3}$ and ~50-100 mmol m$^{-3}$, respectively. In addition, the oxygen concentration is ~50-100 mmol m$^{-3}$ higher in the ACCESS-ESM1.5 across all depths (Garcia et al., 2010a). These biases could result from a strong ocean ventilation or a weak biological pump in the SO. Further details on model performance and evaluation can be found in Ziehn et al. (2020).

For the LIG, a model-data comparison as well as a discussion of the performance of the ACCESS-ESM1.5 compared to other LIG PMIP4 simulations are presented in Yeung et al. (2021), Otto-Bliesner et al. (2021) and Kageyama et al., (2021). A manuscript focusing on Southern Ocean dynamics at the LIG is also in preparation. We have now mentioned this in Section 3.1:

As a result of the insolation anomalies and associated feedbacks, the global mean annual SST anomaly at the LIG compared to PI as simulated by the ACCESS-ESM1.5 equals to 0.17°C, with a pronounced warming at high latitudes and maximum SST warming over the North Atlantic (up to 4°C, Yeung et al. (2021)). Our simulated temperatures are in line with the range of PMIP4 lig127k simulations (Otto-Bliesner et al., 2021). A model-data comparison of the LIG climate state is presented in Yeung et al. (2021).

We have also mentioned the biases in our discussion, e.g. in Section 4:

> Although, the simulated nutrient and alkalinity concentrations are underestimated over the SO, the vertical gradients of these are captured reasonably well (Ziehn et al., 2020). As such, these biases should not significantly impact our results.

3. *I am not sure how the inference of future carbon cycle change would be useful based on the simulated steady-state response during the LIG. What are the logics behind the linkage?*

A. Both reviewers raise a valid concern regarding the inference of carbon cycle at the LIG for the future. We have now made significant changes to the manuscript to clarify the difference between our LIG simulation and future projections. However, since our simulated Southern Ocean SSTs are higher than PI and sea-ice cover is reduced by up to 41%, we believe that our study adds substantial knowledge to the carbon cycle response to generally warmer conditions.

    i.    Abstract:

> The projected future strengthening and poleward shift of the SH westerlies coupled to warmer conditions at the surface of the SO should thus weaken the capacity of the SO to absorb anthropogenic $CO_2$ over the coming century.

    ii.    Introduction in p1-2 L. 19-23:

> The SO $CO_2$ uptake weakened during the 1990s due to a strengthening and poleward shift of the Southern Hemisphere (SH) mid-latitude westerlies (Lovenduski et al., 2007; Le Quéré et al., 2007; Zickfeld et al., 2007, Gruber et al., 2019a, b), but strengthened in the 2000s due to cooling over the Pacific and increased stratification over the Atlantic and Indian sectors of the SO (Landschuetzer et al., 2016; Gruber et al., 2019b).

    iii.    Introduction in p2, L. 28-29:

> At the same time, SH westerlies are projected to strengthen and shift poleward over the coming century (Collins et al., 2013; Zheng et al., 2013; Goyal et al., 2021).

    iv.    We have removed the comparison with future simulations in P9, L263-264 in the Discussion.

    v.    We have also reworded the concluding statements to explicitly clarify that the LIG is not a good analogue for future changes. The concluding paragraph in the Discussion (P10, L. 302-306) now reads:

> SH westerlies are projected to strengthen and shift poleward over the coming century (Collins et al., 2013; Zheng et al., 2013; Goyal et al., 2021), contrary to the LIG simulations presented here. Thus, changes in the carbon cycle simulated at the LIG may not serve as a

good analogue for potential future changes. Nevertheless, the simulated enhanced SO $CO_2$ outgassing, despite a slight equatorward shift of the westerlies, support a weaker capability of the Southern Ocean to take up anthropogenic $CO_2$ over the coming century.

4. *The authors focused on the Southern Ocean carbon cycle responses to a global scale climate change. In such a steady-state model setup, the globally integrated air-sea CO2 exchange should be close to zero unless there are imbalances caused by riverine input and sedimentary burial. The excess CO2 outgassed through the Southern Ocean needs to be taken up elsewhere. Where and how does this uptake occur? A discussion of this would be useful.*

A. Our results show that while the SO is associated with an increase in outgassing, the largest source of uptake is the North Pacific subpolar gyre (Figure A2 a). This marked uptake is also noticed in the zonal mean uptake between 40-60˚N (Figure A2 b). Large regions in the South Atlantic and the Indian Oceans also show increased uptake, while the North Atlantic is associated with increased outgassing (Figure A2 a). This outgassing over the North Atlantic, and uptake over the South Atlantic and Indian Oceans are broadly consistent with the warming and cooling patterns over the surface of these oceans (Figure R1 and Yeung et al, 2021).

[Figure]

**Figure A2.** (a) Annual mean air-sea $CO_2$ flux (molCm$^{-2}$y$^{-1}$) for LIG-PI. Red colors indicate increased outgassing from the ocean and blue uptake by the ocean. Thick black lines show the zero-line contour. (b) Zonal mean $CO_2$ flux (molCm$^{-2}$y$^{-1}$) for PI (blue) and LIG (red).

[Figure]

**Figure R1**: Annual mean sea surface temperature anomalies with proxy data (filled markers: squares for the compilation by Capron et al., 2014, 2017; dots for the compilation by Hoffman et al., 2017), and contours of annual sea-ice concentration at 15 % overlaid (blue: LIG; yellow: PI). Figure from Yeung, N. K.-H., Menviel, L., Meissner, K. J., Taschetto, A. S., Ziehn, T., and Chamberlain, M.: Land–sea temperature contrasts at the Last Interglacial and their impact on the hydrological cycle, Climate of the Past, 17, 869–885, 2021.

We have now included this in Section 3.2, which now reads:

> This increased outgassing over the SO is compensated by increased uptake over other ocean basins, especially the North Pacific subpolar gyre, the South Atlantic and the northwest Indian Ocean, while higher outgassing is simulated in the North Atlantic (Fig. A2). In this paper, we focus primarily on the SO. To better understand the SO changes, we decompose the oceanic pCO$_2$ changes into their different components.

**Specific comments:**
*Line 10: "changes in sea ice exert a minor control" Is this also true on regional scales?*

A. While this is true for most locations, certain areas around the Weddell and Lazarev Seas could be affected by changes in both sea ice and SST. We have now added this in the discussions in Section 4:

> Reduced solubility due to higher SSTs leads to an increase in outgassing over most of the SO, while the reduced sea-ice cover does not significantly impact the CO$_2$ fluxes. Assessing the impact of sea-ice changes on CO$_2$ fluxes (Appendix A), we find that reduced sea-ice concentration at the LIG in the Weddell and Ross Seas leads to a 5% increase in CO$_2$ uptake in autumn and winter (Fig. A5).

We have also estimated the effect of sea ice changes on the CO2 flux based on Wanninkkhof (2014) and presented the equation and results in the Appendix:

**Appendix A: Effect of sea ice on air-sea gas exchange**

To estimate the effect of sea ice changes at the LIG on the air-sea gas exchange, we use a modified version of the equation of $CO_2$ flux from Wanninkhof (2014):

$$F_{CO2} = 7.7 \times 10^{-4} \times |U^2| \times \Delta pCO_2 \times (1 - Sc) \tag{A1}$$

where $F_{CO2}$ is the $CO_2$ flux ($molCm^{-2}y^{-1}$), U is the average wind speed ($ms^{-1}$), $\Delta pCO_2$ is the difference in partial pressure of $CO_2$ between ocean surface and atmosphere (µatm) and Sc is the sea ice concentration. To investigate the effect of LIG sea ice changes on the LIG $CO_2$ flux, we estimate $F_{CO2}$ using both LIG Sc as well as PI Sc, while keeping all the other variables at LIG values. The resulting patterns are presented in Fig. A5. Figure A5 shows that the reduced sea ice during LIG compared to PI leads to less than 5% increase in carbon sink over the Ross Sea region all year round and Weddell Sea region over autumn and winter, and around 2% increase in outgassing in the Lazarev and Weddell Seas.

[Figure]

**Figure A5**: Calculated seasonal $CO_2$ fluxes using LIG $\Delta pCO_2$, winds and (left) LIG sea-ice concentration, (middle) PI sea ice concentrations; (right) differences in $CO_2$ flux for calculations using the LIG sea ice concentrations and compared to PI sea ice concentrations. Notice the reduced color scale in the third column. Calculations are based on Wanninkhof (1992) and Wanninkhof (2014). The red and blue contours indicate 15% sea ice concentrations for LIG and PI respectively.

Additionally, reduced sea ice cover could also have allowed for increase in Net Primary Production (NPP). We have now amended Section 3.4:

> The changes in SST and upwelling (Fig. 1), alongside the reduced sea ice extent, lead to an increase in net primary production (NPP) and export production over the SO by ~17% and ~11% respectively (Fig. A2a and b).

*Line 11-12: what are the logics behind the linkage between the simulated LIG steady-state response and future transient responses in the coming century?*

A. Our simulated Southern Ocean SSTs at the LIG are higher than PI and sea-ice cover is reduced by up to 41%. We therefore believe that our study adds substantial knowledge to the carbon cycle response to generally warmer conditions. We have amended the manuscript to better highlight the differences between our LIG simulation and future projections, and the inferences from our study that can inform future responses. These are mentioned earlier, in response to comment 3.

*Line 18: 25% each or together?*

A. We have now clarified this. The text now reads:

> Both land and ocean presently act as sinks of anthropogenic carbon, each absorbing about 25% of anthropogenic emissions (Friedlingstein et al., 2020), with 40% of the ocean sink being attributed to the SO (Caldeira and Duffy, 2000; DeVries, 2014).

*Line 100: 275 ppm for radiative forcing only? The authors earlier stated that they used 280 ppm.*

A. The piControl simulation is forced with a $CO_2$ value of 284.3 ppm while the lig127k experiment is forced with a $CO_2$ of 275 ppm, in accordance with the CMIP6 and PMIP4 protocols. Alongside DIC corresponding to these prescribed $CO_2$ values, our modelling setup contains an additional tracer of DIC corresponding to a 280 ppm $CO_2$ values. This enables us to assess the effects of the climate on the carbon cycle independently of the background atmospheric $CO_2$ concentration. We use the 280 ppm based tracer for all of our analyses in the paper to quantify the effects of the LIG climate on the carbon cycle. We have now amended the text to make this distinction clearer. After discussing the experiment protocols and forcing scenarios relevant to the piControl and lig127k experiments, we have added the description of our multiple tracers in Section 2.1:

> WOMBAT includes two DIC tracers, one being forced by the prescribed atmospheric $CO_2$ concentration (PI: 284.3 ppm, LIG: 275 ppm), and the other forced by a constant atmospheric $CO_2$ concentration of 280 ppm. Unless mentioned otherwise, all of the analyses presented here are based on the tracers forced with $CO_2$ concentrations of 280 ppm, while the climate

response is forced with the radiative forcing of 284.3 ppm and 275 ppm, respectively. This allows quantification of the effects of the LIG climate on the carbon cycle independently of the difference in background $CO_2$ concentrations.

*Line 110 and elsewhere: To my best understanding, the Revelle factor refers to a sensitivity of pCO2 with respect to DIC only, not to SST, SSS and ALK.*

A. The reviewer correctly points out that the Revelle factor refers to the sensitivity of $pCO_2$ to DIC only. We have now corrected the text throughout the manuscript to only refer the DIC sensitivity as 'Revelle factor'. Dependence on the other variables, SST, SST and ALK are now correctly referred to as 'sensitivities'. This is amended in Section 2.2:

$\gamma_{SST} = 0.0423°C^{-1}$ is the sensitivity of $pCO_2$ to changes in SST (Sarmiento and Gruber, 2006).

And in

$\gamma_X$ is the sensitivity of $pCO_2$ to changes in variable 'X' ($\gamma_{DIC}$ is known as the Revelle factor).

*Line 169: could the increased uptake south of 62S be due to reduced sea ice? Reduced convection in the bottom water formation regions should decrease CO2 uptake.*

A. The reviewer rightly points out that convection may not sufficiently explain the increase in $CO_2$ uptake, and changes in sea ice may play a role. Our analysis suggests the reduced convection in this area reduces mixing with deeper waters that have higher DIC concentrations. In addition, this area of enhanced $CO_2$ uptake corresponds to sea-ice free conditions at the LIG, whereas this region was covered by sea-ice in winter in our PI simulation. Reduced winter sea-ice could enhance $CO_2$ uptake. We have now clarified this in Section 3.2, lines 195-197:

The zonal mean $CO_2$ flux in Fig. 2d shows a small increase in uptake south of 62°S, possibly due to reduced mixing and reduced winter sea ice cover.

*Line 188-202: The net change in pCO2 is really small at 1.2 uatm due to compensating effects of SST and DIC+ALK on pCO2 change. Perhaps this might be smaller than measurement or PI pCO2 uncertainties? When averaged over the Southern Ocean, it is true that SST is a dominant factor. However, the dominating factors would be different regionally because the contributions from SST, DIC, and ALK seem to be highly variable in space. The authors discussed the zonally averaged contributions, but the contributions seem to be also variable in a longitudinal direction as well. Would it be useful to provide a spatial map to identify the dominant factor at each grid cell? For example, SST dominance in red, DIC dominance in yellow, ALK dominance in blue, and SSS dominance in green?*

A. Following the reviewer's suggestion, we have now created a map showing the dominant contributors to pCO$_2$ per grid point. Since DIC and ALK contributions are significantly higher than those from SST and SSS, we have compared the SST+SSS component (solubility) with the DIC+ALK component. This is presented in Figure R2.

[Figure]

**Figure R2**: Dominant contributor to the pCO$_2$ decomposition for (a) global and (b) Southern Ocean. Red colors indicate the solubility component has the higher magnitude, while blue colors indicate the DIC+ALK components to have higher magnitudes.

Following their recommendation, we have now replaced the pCO$_2$ contribution from SSS with that from DIC+ALK in Figure 3d to better reflect the relative effects of SST and DIC+ALK. The pCO$_2$ contribution from SSS is now available in the appendix Figure A3. We believe that Figure R2 above does not add much information in addition to the new Figure 3, and have thus decided to not include it in the manuscript.

[Figure]

**Figure 3.** Attribution of changes in annual mean surface partial pressure of $CO_2$ ($pCO_2$) over SO (µatm). (a) Summary of decomposition of $pCO_2$ over SO. Simulated difference in $pCO_2$ (black circle) and sum of contributions from all components (gray square, Section

2.2). pCO2 change from solubility (SST+SSS, red triangle), and sum of DIC and ALK components (blue diamond). Individual contributions from SST (brown triangle), SSS (magenta triangle), DIC (cyan diamond), and ALK (green diamond). (b) Map of the sum of $pCO_2$ contributions from all components (corresponding to gray square in (a)) in shading, overlaid with the $pCO_2$ change simulated by the model as contours (corresponding to black circle in (a)). Maps of individual $pCO_2$ contributions from (c) SST (corresponding to the brown triangle in (a)), (d) DIC and ALK (corresponding to the blue diamond in (a)), (e) DIC (corresponding to the cyan diamond in (a)), and (f) ALK (corresponding to the green diamond in (a)). (g) Zonal mean contributions to $pCO_2$ over the SO (µatm) for simulated $\Delta pCO_2$ (black), sum of contributions from all components (gray), solubility (red), sum of DIC and ALK components (blue), SST (dashed brown), SSS (dashed magenta), DIC (dashed cyan), and ALK changes (dashed green). Note the non-linear scale on the top and bottom thirds of (g).

*Line 206-207: negative? The red line seems to suggest positive values south of 75S in Fig. 3g.*

A.  We have now removed this sentence.

*Line 218: It is interesting to see the larger SST increases over the Southern Ocean compared to other ocean surfaces, which is responsible for the net increase in oceanic pCO2 over the Southern Ocean during the LIG compared to the PI. Perhaps, it would be useful to discuss why the SST increase is larger over the Southern Ocean than the global average during the LIG?*

A.  Our lig127k simulation shows positive SST anomalies over most of the Northern Hemisphere and the Southern Hemisphere extratropics. This is because of the differences in seasonal insolation patterns. The sea surface warming is highest over the North Atlantic (+ 4°C, Figure 3c of Yeung et al., 2021).

    We have updated this in the discussion of SST changes in Section 3.1, which now reads:

> As a result of the insolation anomalies and associated feedbacks, the global mean annual SST anomaly at the LIG compared to PI as simulated by the ACCESS-ESM1.5 equals to 0.17°C, with a pronounced warming at high latitudes and maximum SST warming over the North Atlantic (up to 4°C, Yeung et al. (2021)). Our simulated temperatures are in line with the range of PMIP4 lig127k simulations (Otto-Bliesner et al., 2021). A model-data comparison of the LIG climate state is presented in Yeung et al. (2021).
>
> A mean 0.53°C warming is simulated over the SO south of 40°S. Warmer conditions are simulated everywhere south of 50°S apart from a ~1°C cooling centered at 58°S in the South Atlantic, and up to a 1.5°C cooling in the subantarctic eastern Pacific. The strongest warming is simulated over the southeast Atlantic and Indian Ocean sectors (up to 3°C). Regional SSTs up to 4°C higher are simulated around 60°S for both austral spring (Fig. A1b) and summer (not shown). The higher SSTs over the SO compared to PI are accompanied by a marked

reduction in sea-ice extent over both austral summer and winter (Fig. 1a), peaking at 41%
reduction in austral winter (Fig. A1a).

*Line 228-230: why does the northward shift of AAIW lead to lower O2 and higher DIC and
PO4? If it is the shift that causes the changes, we might expect some dipole patterns?*

A. The reviewer rightly points out that just a shift of AAIW would not induce these changes.
   We find that along with a northward shift of AAIW, there is also a slowdown of the AAIW
   formation rates due to the northward shift and weakening of the winds (Fig 1e). This leads
   to increased residence times of intermediate waters, and therefore a reduction in dissolved
   oxygen and an increase in remineralized carbon at intermediate depths of the SO north of
   55˚S (Figure 4d). We have now updated the text to clarify this in Section 3.4:

This leads to ~10% weaker upwelling south of 55˚S and up to ~20% stronger upwelling
north of 55˚S. Due to a northward shift and weakening of winds (Fig. 1e, f), the Antarctic
Intermediate Water (AAIW) formation regions shift northward and the AAIW formation rate
slows down (Downes et al., 2017).

   And:

These circulation changes impact the DIC distribution in the ocean (Fig. 4b). For instance,
the reduced formation rate of AAIW (Fig. 1d) increases residence times and leads to lower
dissolved oxygen (Fig. 4a), higher PO4 (Fig. 4c), higher remineralized carbon (Fig. 4d) and
higher DIC concentrations (Fig. 4b) at intermediate depths of the SO north of 55˚S.

*Line 283: The role of sea ice cover change can be tested offline. You can calculate air-sea CO2
fluxes based the simulated air-sea pCO2 difference with and without the sea ice cover changes.*

A. The reviewer rightly points out that the effect of sea ice changes can be tested offline. We
   have now performed this calculation based on Wanninkhof (1992) and Wanninkhof (2014),
   and found that reduced LIG sea-ice cover only leads to minor changes next to the Antarctic
   coast. Figure A5 shows that the reduced sea ice during LIG compared to PI leads to less
   than 5% increase in carbon sink over the Ross Sea region all year round and Weddell Sea
   region over autumn and winter, and around 2% increase in outgassing in the Lazarev and
   Weddell Seas.

[Figure]

**Figure A5**: Calculated seasonal $CO_2$ fluxes using LIG $\Delta pCO_2$, winds and (left) LIG sea-ice concentration, (middle) PI sea ice concentrations; (right) differences in $CO_2$ flux for calculations using the LIG sea ice concentrations and compared to PI sea ice concentrations. Notice the reduced color scale in the third column. Calculations are based on Wanninkhof (1992) and Wanninkhof (2014). The red and blue contours indicate 15% sea ice concentrations for LIG and PI respectively.

We have now mentioned this in the Discussion section:

> Reduced solubility due to higher SSTs leads to an increase in outgassing over most of the SO, while the reduced sea-ice cover does not significantly impact the $CO_2$ fluxes. Assessing the impact of sea-ice changes on $CO_2$ fluxes (Appendix A), we find that reduced sea-ice concentration at the LIG in the Weddell and Ross Seas leads to a 5% increase in $CO_2$ uptake in autumn and winter (Fig. A5).

And added the calculations in the Appendix:

> **Appendix A: Effect of sea ice on air-sea gas exchange**
>
> To estimate the effect of sea ice changes at the LIG on the air-sea gas exchange, we use a modified version of the equation of $CO_2$ flux from Wanninkhof (2014):
>
> $$F_{CO2} = 7.7 \times 10^{-4} \times |U^2| \times \Delta pCO_2 \times (1 - Sc) \tag{A1}$$

where $F_{CO2}$ is the $CO_2$ flux ($molCm^{-2}y^{-1}$), U is the average wind speed ($ms^{-1}$), $\Delta p\ CO_2$ is the difference in partial pressure of $CO_2$ between ocean surface and atmosphere ($\mu atm$) and Sc is the sea ice concentration. To investigate the effect of LIG sea ice changes on the LIG $CO_2$ flux, we estimate $F_{CO2}$ using both LIG Sc as well as PI Sc, while keeping all the other variables at LIG values. The resulting patterns are presented in Fig. A5. Figure A5 shows that the reduced sea ice during LIG compared to PI leads to less than 5% increase in carbon sink over the Ross Sea region all year round and Weddell Sea region over autumn and winter, and around 2% increase in outgassing in the Lazarev and Weddell Seas.

*Line 301: There is a study suggesting an increase in surface productivity over the Southern Ocean in a warming climate (e.g., Kwiatkowski et al. (2020)).*

A. We thank the reviewer for this suggestion, and have now included this reference in the discussions:

SH westerlies are projected to strengthen and shift poleward over the coming century (Collins et al., 2013; Zheng et al., 2013; Goyal et al., 2021), contrary to the LIG simulations presented here. Also, while we find a more efficient biological pump at the LIG, Boyd (2015) suggested the biological pump to be less efficient under a future warming scenario, although recent studies have suggested a possible increase in surface productivity in the future (Kwiatkowski et al., 2020).

*Figures: please add latitude and longitude labels to maps*

A. This was fixed, thank you.